# Bioactive Compounds in *Moringa oleifera*: Mechanisms of Action, Focus on Their Anti-Inflammatory Properties

**DOI:** 10.3390/plants13010020

**Published:** 2023-12-20

**Authors:** Adina Chiș, Paul Aimé Noubissi, Oana-Lelia Pop, Carmen Ioana Mureșan, Michel Archange Fokam Tagne, René Kamgang, Adriana Fodor, Adela-Viviana Sitar-Tăut, Angela Cozma, Olga Hilda Orășan, Simona Codruța Hegheș, Romana Vulturar, Ramona Suharoschi

**Affiliations:** 1Department of Molecular Sciences, “Iuliu Hațieganu” University of Medicine and Pharmacy, 6 Louis Pasteur St, 400349 Cluj-Napoca, Romania; adinachis82@gmail.com (A.C.); romanavulturar@gmail.com (R.V.); 2Department of Animal Biology and Conservation, Faculty of Science, University of Buea, Buea P.O. Box 63, Cameroon; noubiaime@yahoo.fr (P.A.N.); gemskruy@yahoo.fr (R.K.); 3Department of Food Science, University of Agricultural Science and Veterinary Medicine, 3-5 Calea Mănăștur, 400372 Cluj-Napoca, Romania; oana.pop@usamvcluj.ro (O.-L.P.); ramona.suharoschi@usamvcluj.ro (R.S.); 4Molecular Nutrition and Proteomics Lab, CDS3, Life Science Institute, University of Agricultural Science and Veterinary Medicine, 3-5 Calea Mănăștur, 400372 Cluj-Napoca, Romania; 5Department of Biological Sciences, Faculty of Science, University of Ngaoundéré, Ngaoundéré P.O. Box 454, Cameroon; fm_archange@yahoo.fr; 6Clinical Center of Diabetes, Nutrition and Metabolic Diseases, “Iuliu Hațieganu” University of Medicine and Pharmacy, 2-4 Clinicilor St., 400012 Cluj-Napoca, Romania; adriana.fodor@umfcluj.ro; 7Department of Internal Medicine, Faculty of Medicine, “Iuliu Hațieganu” University of Medicine and Pharmacy, 400012 Cluj-Napoca, Romania; adelasitar@yahoo.com (A.-V.S.-T.); angelacozma@yahoo.com (A.C.); olgaorasan@gmail.com (O.H.O.); 8Department of Drug Analysis, Faculty of Pharmacy, “Iuliu Hațieganu” University of Medicine and Pharmacy, Louis Pasteur Street 6, 400349 Cluj-Napoca, Romania

**Keywords:** *Moringa oleifera*, anti-inflammatory activity, NAFLD, quercetin, kaempferol, chronic inflammatory diseases, anti-oxidative stress, type 2 diabetes, nuclear factor-kappa B, polyphenols

## Abstract

*Moringa oleifera* (*M. oleifera*) is a tropical tree native to Pakistan, India, Bangladesh, and Afghanistan; it is cultivated for its nutritious leaves, pods, and seeds. This scientific study was conducted to outline the anti-inflammatory properties and mechanisms of action of bioactive compounds from *M. oleifera.* The existing research has found that the plant is used in traditional medicine due to its bioactive compounds, including phytochemicals: flavonoids and polyphenols. The compounds are thought to exert their anti-inflammatory effects due to: (1) inhibition of pro-inflammatory enzymes: quercetin and kaempferol inhibit the pro-inflammatory enzymes (cyclooxygenase and lipoxygenase); (2) regulation of cytokine production: isothiocyanates modulate signaling pathways involved in inflammation, such as the nuclear factor-kappa B (NF-kappa B) pathway; isothiocyanates inhibit the production of pro-inflammatory cytokines such as TNF-α (tumor necrosis factor α) and IL-1β (interleukin-1β); and (3) antioxidant activity: *M. oleifera* contains flavonoids, polyphenols, known to reduce oxidative stress and inflammation. The review includes *M. oleifera*’s effects on cardiovascular protection, anti-hypertensive activities, type 2 diabetes, inflammatory bowel disease, and non-alcoholic fatty liver disease (NAFLD). This research could prove valuable for exploring the pharmacological potential of *M. oleifera* and contributing to the prospects of developing effective medicines for the benefit of human health.

## 1. Introduction

The wide range of therapeutic effects exhibited by plant-based bioactive substances has positioned them as promising candidates in contemporary drug development; many phytochemicals have shown antibacterial, antioxidant, anticancer, anti-hepatitis C virus, and anti-inflammatory potentials such as quercetin, curcumin, capsaicin, resveratrol, taxol, and others; in addition, plant-based natural products have shown their therapeutic potential against various cardiovascular diseases or brain and neurodegenerative diseases (such as Parkinson’s disease, Alzheimer’s disease, and Huntington’s disease) [1].

Moringa stands as the sole genus in the Moringaceae family of flowering plants [2,3]. Among its various species, *M. oleifera* Lam (*Moringa pterygosperma* G.) is the most popular and widespread and is native to Pakistan, India, Bangladesh, and Afghanistan. It was found to withstand even the driest and harshest of soils [4]. It is largely cultivated in subtropical and tropical areas, with its young leaves, seed pods, and mature seeds serving as vegetables for populations in many countries [5], especially in the northern part of Cameroon. Also known as “horseradish tree”, or “drumstick tree”, all parts of *M. oleifera* have long been consumed by humans for nutritional and medicinal purposes [5]. Due to its medicinal properties, the plant was sometimes named “mother’s best friend” or “miracle tree” [6]. A recent article showed that the supplementation of polar extracts of *M. oleifera* was highly effective in controlling oxidative stress, inducing the retrieval of sensory and motor functions, and therefore facilitating accelerated nerve generation [7]. In a recent study, the phenolic composition and antioxidant capacity of various dietary supplements derived from *M. oleifera* were examined. The study found an important correlation, revealing that greater phenolic content corresponded to increased antioxidant activity within these supplements, regardless of their different forms of presentation [8].

Recently, it has been demonstrated that *M. oleifera* leaf extract alleviates hepatotoxicity caused by antiretroviral drugs through its ability to neutralize harmful oxidants and activate the NRF2 antioxidant pathway. This highlights *M. oleifera*’s substantial therapeutic promise and suggests its potential role as a valuable supplement for mitigating the toxicity associated with antiretroviral drugs [9].

Regarding the fortified foods, the yogurts fortified with *M. oleifera* exhibited superior antioxidant properties compared to the negative control; these results underscore the potential utilization of *M. oleifera* powder and extract as natural supplements for creating fortified foods that may help address malnutrition [10]. In addition, a recent publication has shown the development of a palatable and standardized pharmacologically active formulation using *M. oleifera* leaves, designed as a functional food. This formulation aims to activate NRF2 signaling and can be consumed either as a beverage (such as hot soup) or in the form of a freeze-dried powder. Its goal is to reduce the risk of environmental respiratory diseases by harnessing the power of isothiocyanate moringin and polyphenols, both strong stimulators of NRF2 signaling [11].

Additionally, recent studies are outlining the interest in enhancing the viable bioactive compounds in the culture of *M. oleifera*, i.e., the in vitro-based elicitation approach (a biotechnological tool for enhancing the production of secondary metabolites) and demonstrating that utilizing varying spectral lights represents an effective method for increasing the production of nutraceuticals and novel pharmacologically significant metabolites in the in vitro callus culture of *M. oleifera* [12]. The high nutritional, nutraceutical, and therapeutic profile (Table 1) is mainly attributed to its rich repertoire of biologically active molecules: proteins (peptides and protein hydrolysates), flavonoids, saponins, phenolic acids, tannins, isothiocyanates, lipids, minerals, and vitamins, amongst others [13].

The protein quality of *M. oleifera* leaves was shown to be similar to that of milk and eggs [36], and the plant has been described as performing numerous pharmacological properties and is long known in Ayurvedic medicine. Many recent studies have proven that leaves, pods, seeds, flowers, roots, bark, and stem has anti-inflammatory effects (Figure 1) [2,37,38,39,40,41,42,43,44].

*M. oleifera* was reported to be richer in potassium than bananas, with a higher content of calcium compared to milk, more iron than spinach, more vitamin C than oranges, and a higher vitamin A content compared to carrots [45].

**Figure 1 plants-13-00020-f001:**
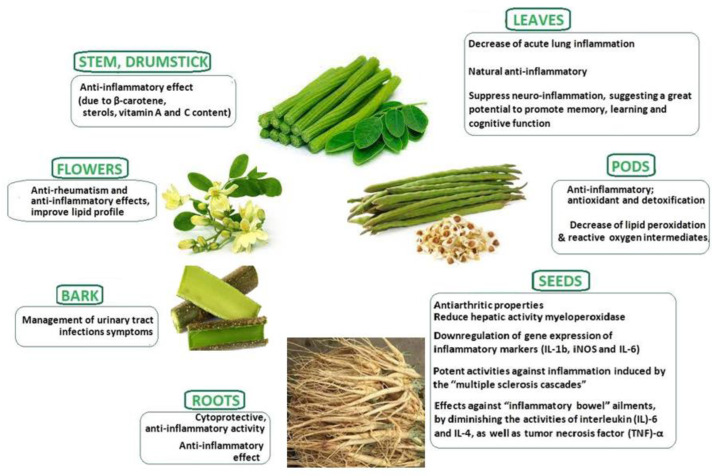
Summary of pharmacological (direct and indirect) anti-inflammatory actions of different parts of *M. oleifera* (leaves, pods, seeds, flowers, roots, bark, and stem) demonstrated on an animal model (based on details cited in references [2,13,20,32,34,37,38,39,40,41,42,43,44,46]).

This review aims to provide detailed information on *M. oleifera* anti-inflammatory compounds and their possible specific mechanisms of action.

## 2. Description of *Moringa oleifera*

The genus is composed of about 13 species, namely *M. arborea* Verdcourt (Kenya, Somalia), *M. borziana* Mattei (Kenya and Somalia), *M. concanensis* Nimmo (India), *M. drouhardii* Jumelle (Southern Madagascar), *M. hildebrandtii* Engler (Southwest Madagascar), *M. longituba* Engler (Kenya, Southeast Ethiopia, and Somalia), *M. oleifera* Lam. (Cameroun and India), *M. ovalifolia* Dinter ex Berger (Namibia and Southwest Angola), *M. peregrina* Forssk. Ex Fiori (Red Sea, Arabia, and Northeast Africa), *M. pygmaea* Verdcourt (North Somalia), *M. rivae* Chiovenda (Kenya and Ethiopia), *M. ruspoliana* Engler (Kenya, Ethiopia, and Somalia), and *M. stenopetala* (Baker f.) Cufodontis (Kenya, Southwest Ehiopia, and Somalia), is distributed in Southwest Asia, Africa, and Madagascar [47].

Moringa tree has a diameter of about 45 cm (1.5 ft) and is about 10–12 m (32–40 ft) in height. Its fragrant bisexual flowers are surrounded by five thinly and unequally veined yellow to white petals of about 1.0 to 1.5 cm long and broad of about 2.0 cm. Fruits referred to as pods are tri-lobed pendulous capsules that are brown and triangular and split lengthwise into three parts when dried. Fruits contain about 26 seeds and are mostly produced between March and April. Immature pods are green in color; they turn brown at maturity. Seeds are around 1 cm in diameter and range in color from brown to black but can also be white when the viability of the kernels is low. Seeds germinate in 2 weeks, while each tree can produce around 15,000 to 25,000 seeds/year. The average weight is 0.3 g/seed. Propagation is usually performed by wind and water [36].

Taxonomically, *M. oleifera* is a member of the Kingdom of Plantae; sub kingdom of Tracheobionta; super-division of Spermatophyta; the division of Magnoliophyta; class of Magnoliopsida; subclass of Dilleniidae; order of Capparales; family of Moringceae; genus Moringa; and specie, oleifera [36,48].

## 3. Bioactive Components of *M. oleifera* and Their Mechanisms of Action in Inflammatory Diseases

In recent years, *M. oleifera*, a versatile and nutritionally rich plant, has garnered significant attention for its potential therapeutic benefits in combating inflammatory diseases. The description of bioactive compounds found in *M. oleifera* and the approach to intricate mechanisms of action are important in addressing inflammatory conditions, providing promising insights for future health interventions.

### 3.1. M. oleifera Essential Oil and Fatty Acids

The phytochemical composition of the volatile constituents of the ethanolic extract from the leaves of *M. oleifera* was reported by Kute in 2017; a total of forty-four volatile compounds have been identified with pentacosane (**1**) (17.4%), hexacosane (**2**) (11.2%), (E)-phytol (**3**) (7.7%), and 1-[2,3,6-trimethylphenyl]-2-butanone (**4**) (3.4%) as major constituents [2]. *M. oleifera* seed oil content ranges between 35 and 40% [49]. Oleic acid (**5**) (65%), linoleic acid (**6**) (16%), palmitic acid (**7**) (12.31%), stearic acid (**8**) (5.1%), and palmitoleic acid (**9**) (2.1%) are found to be its most important fatty acids [50] (Figure 2).

### 3.2. Bioactive Phytochemical Components

In the strictest sense, phytochemicals are the chemicals generated by plants. They encompass a broad group of compounds, known as secondary metabolites, that naturally occur and accumulate in plants at high concentrations [51]. Phytochemicals are found in roots, seeds, leaves, stems, flowers, and pods [52]. According to their chemical structure and characteristics, plant phytochemicals are classified into five main groups: polyphenols, sulfur-containing compounds, carotenoids, alkaloids, and terpenoids [52,53].

Polyphenols are classified as phenolic acids (with only one phenol ring present) and flavonoids (with more than one phenol ring present) [53] (Figure 3). Phenolic acids and flavonoids (especially tannins) are among the most important polyphenols found in the Moringa tree [53]. Leaves were reported to have the highest total phenolic contents, with about 2000 to 12200 mg GAE/100 g [53]. Except for the roots and seeds, the most common flavonoids found in various parts of the Moringa tree are kaempferol glycosides [glucosides (**11**), malonyl glucosides (**12**), and rutinosides (**13**)] and quercetin (**14**) [53]; other flavonols found in lower amounts include myricetin (**15**), epicatechin (**16**), and rutin (**17**). Among the abundant phenolic acids in the Moringa tree are caffeic acid (**18**), chlorogenic acid (**19**), coumaric acid (**20**), gallic acid (**21**), and ellagic acid (**22**) [53]. Leaves of Moringa were also reported to contain an important amount of tannins whose concentration can vary, with the highest being reported in dried leaves. A low quantity of tannins was also reported in seeds [53]. 

*M. oleifera* contains various sulfur compounds, including glucosinolates and isothiocyanates (Figure 4). When the plant is damaged, processed, harvested, or chewed, the enzyme called myrosinase is activated. This enzyme catalyzes the hydrolysis of β-D-glucose at neutral pH, resulting in the formation of isothiocyanates, thiocyanates, sulfates, and nitriles [53].

Glucosinolates represent a diverse category of glycosidic compounds containing sulfur and nitrogen, found in abundance in numerous plant species. Various types of glucosinolates have been identified in different parts of the Moringa plant, including the leaves, pods, stem, and roots. The most common glucosinolate in this plant is the 4-(α-l-rhamnopyranosiloxy) benzyl glucosinolate called glucomoringin (**23**). Glucomoringin (**23**) is commonly present in stems, flowers, pods, leaves, and seeds, while in the roots the predominant glucosinolate is benzyl glucosinolate (**24**). These compounds contribute to the pungent taste and odor of the plant and exhibit biological properties such as antifungal and antibacterial effects [54]. 

Isothiocyanates with thiocyanates and thiocarbamates are secondary metabolites derived from glucosinolates and have garnered significant attention because of their pharmacological properties. The principal isothiocyanates isolated from *M. oleifera* include niazidin (**25**), niazicin (**26**), or niazinin (**27**) [54]. 

Carotenoids are highly unsaturated fat-soluble pigments that provide fruits, vegetable fungi, bacteria, and algae their characteristic red, orange, or yellow color [55,56] (Figure 5). Carotenoids are subdivided into two groups: the carotenes (precursor of vitamin A) with a single long carbon chain and the xanthophylls, with oxygen atoms attached to their structure [56]. Moringa fresh leaves were found to contain an important amount of β-carotene (6.6–17.4 mg/100 g), higher than in carrots, pumpkins, and apricots [55,57,58]. The β-carotene content in Moringa dried leaves was even higher, about 23.31 to 39.6 mg per 100 g of dry matter. Other different carotenoids were identified in the flowers, fruits, and fresh leaves of eight Moringa oleifera commercially grown cultivars in India and included all-E-β-carotene (**28**), all-E-zeaxanthin (**29**), all-E-lutein (**30**), all-E-luteoxanthin (**31**), 15-Z-β-carotene (**32**), and 13-Z-lutein (**33**) [55].

Alkaloids are an assembly of naturally occurring chemical composites, typically comprising basic nitrogen atoms; they are miscellaneous elements and biomolecules, secondary compounds derived from amino acids or transamination [59] (Figure 6). There are three major alkaloid groups, namely pseudoalkaloids, true alkaloids, and protoalkaloids. Compared to pseudoalkaloids, the true alkaloids and protoalkaloids are derived from amino acids. Almost all true alkaloids have a bitter taste [59]. The presence of many alkaloids has been reported in the Moringa tree. N,α-l-rhamnopyranosyl vincosamide (**34**) was the most commonly reported Moringa plant indol alkaloid, which was isolated from the leaves. These leaves were also reported to contain glycosides of a pyrol alkaloid such as marumosides A (pyrrolemarumine 4″-O-α-l-rhamnopyranoside) (**35**) and marumosides B (4′-hydroxyphenylethanamide) (**36**) [53]. *M. oleifera* was also reported to contain two, as trigonelline (**37**) [60] or moringinine (benzylamine) (**38**) [61]. 

Terpenoids are compounds synthesized from the condensation of the five-carbon precursor isopentenyl pyrophosphate (IPP) with dimethylallyl pyrophosphate (DMAPP), the functional isomer [62,63] (Figure 7). Under the enzyme isoprenyl diphosphate synthase, IPP and DMAPP are condensed into acyclic and achiral isoprenyl diphosphate/pyrophosphate (ID, C5n) intermediates considered the universal precursors of terpenoid. Terpene synthases (TPSs) action on one or more of these precursors produces a diversity of terpenes [63]. Lupeol acetate (**39**), α-amyrin (**40**), and β-amyrin (**41**) are terpenes isolated from a n-hexane fraction of the ethanol extract of *Moringa peregrina* aerial parts [47]. 

Furthermore, proteins and peptide fractions with a high nutritional profile have been studied as promising components in Moringa [22].

Indeed, these phytochemicals in the Moringa tree are largely believed to be responsible for its diverse biological activities and disease-preventive potential. The presence and amount of these metabolites depend on geographical location, soil type, and climate [64].

### 3.3. Anti-Inflammatory Activities of M. oleifera Compounds and Their Mechanisms of Action

The compounds of *M. oleifera* are thought to exert their anti-inflammatory effects through several mechanisms [39,41]: (a) inhibition of pro-inflammatory enzymes: quercetin (**14**) and kaempferol (**10**) are *M. oleifera*’s compounds that inhibit the activity of pro-inflammatory enzymes [cyclooxygenase (COX) and lipoxygenase (LOX)], which are key enzymes involved in the production of inflammatory mediators such as prostaglandins and leukotrienes; (b) regulation of cytokine production: isothiocyanates (a class of *M. oleifera*’s compounds) have been shown to modulate signaling pathways involved in inflammation, as is the nuclear factor-kappa B (NF-kappa B) pathway; this ensures modulation of signaling pathways. Isothiocyanates have also been shown to inhibit the production of pro-inflammatory cytokines [tumor necrosis factor-α (TNF-α) and interleukin-1β (IL-1β)] and to increase the production of anti-inflammatory cytokines such as interleukin-10 (IL-10); and (c) antioxidant activity: flavonoids and polyphenols help to reduce oxidative stress and inflammation. These compounds may also inhibit the activity of pro-inflammatory enzymes and modulate cytokine production. 

Inflammation constitutes a vital and intricate aspect of an organism’s reaction to biological, chemical, and/or physical stimuli [65]. Inflammation is commonly characterized by distinct acute and chronic phases, although there is some overlap between these stages. During the acute phase, primarily granulocytes, guided by a chemotactic gradient, migrate to the site of injury. This orchestrated response, facilitated by acute phase proteins and cytokines, aims to eliminate the inflammatory stimulus (e.g., infectious agents or foreign material) and remove damaged cells, initiating the healing process [66]. Depending on the severity of the injury, this acute cellular phase could be enough to resolve any damage. As a result of either prolonged exposure to inflammatory stimuli or an inappropriate reaction to self-molecules, persistent inflammation can probably lead to the chronic phase. The active immune cell populations shift to include a mononuclear phenotype, resulting in tissue damage and fibrosis. During inflammation, activated macrophages secrete a certain number of different pro-inflammatory cytokines, including TNF-α, IL-1β, interferon-γ (IFN-γ), interleukin-6 (IL-6), and oxidative stress mediators, such as nitric oxide (NO), produced by iNOS, the inducible nitric oxide synthase [67]. Chronic inflammation is implicated in the pathophysiology of numerous disorders, including cardiovascular diseases (atherosclerosis and hypertension), type 2 diabetes, enterocolitis, and non-alcoholic fatty liver disease (NAFLD). *M. oleifera* extract has shown potential activities against these diseases (see Figure 1) [68]. 

#### 3.3.1. Cardiovascular Protection and Anti-Hypertensive Activities of *M. oleifera*


Chronic inflammatory diseases such as systemic lupus erythematosus, rheumatoid arthritis, psoriasis, and HIV infection affect up to 18% of the global population [69]. They have a higher risk of developing inflammation-related cardiovascular diseases than the general population [69,70,71,72,73]. Inflammation leads to atherosclerosis, ischemic heart disease, and heart failure, partly through atherosclerotic plaque formation. Notwithstanding the considerable enhancements in our insight into the etiology of cardiovascular diseases (CVDs), stroke, coronary artery disease, and other vasculopathies still account for over 31% of all fatalities globally [74]. Although much of these cardiovascular (CV) risks are due to traditionally known cardiovascular disease risk factors such as diabetes, hyperlipidemia, hypertension, and smoking, inflammation has been considered and identified as the key factor in the development, evolution, and aggravation of atherosclerosis [75]. Increased levels of pro-inflammatory factors in the serum, such as IL-6, IL-12, and TNF-α, and inflammatory biomarkers, such as C-reactive protein (CRP), high-sensitivity C-reactive protein (hs-CRP), fibrinogen, and homocysteine YKL-40, have been observed in atherosclerotic patients. YKL-40 has been confirmed to be a key factor in the pathogenesis of cardiovascular diseases such as insulin resistance and obesity. Furthermore, it is highly expressed in atherosclerotic plaques [76]. 

Hypertension is usually characterized by high systolic and diastolic blood pressure resulting from increased arginase, acetylcholinesterase, phosphodiesterase-5, and angiotensin-1-converting enzymes [77]. Hypertension has also been associated with insufficient production of the principal vasodilator, nitric oxide (NO), and vascular remodeling. Endogenous production of nitric oxide by NO synthases (NOS) requires L-arginine as the substrate. L-arginine also serves as a substrate for arginases, metabolizing L-arginine into urea and L-ornithine. It was recently found that arginase activity can result from many hypertensive stimuli, such as persistent chronic inflammation and salt loading. Stimulation of arginase enzymatic activity reduces the bioavailability of L-arginine (to NOS), thereby decreasing endogenous NO production in the vasculature. L-ornithine can be converted into polyamines and proline, metabolites central to vascular remodeling and proliferation of vascular smooth muscle cells. Thus, arginase implication in hypertension pathogenesis promotes vascular remodeling and inhibition of endogenous NO production [78]. The extracts from *M. oleifera* leaves and seeds demonstrated a reduction in activity of the enzyme arginase. These inhibitory effects may be attributed to the actions of phenolic compounds, which have been reported to inhibit the activity of the arginase enzyme [77]. Ethanolic extracts from the leaves of *M. oleifera* showed notable anti-hypertensive or hypotensive activity [47]. Thiocarbamates such as 4-[(4′-*O*-acetyl-α-l-rhamnosyloxy)benzyl]isothiocyanate, 4-[(α-l-rhamnosyloxy)benzyl] isothiocyanate, niazinin A (**27**), niazicin A (**26**), and niazirin [79] have been identified and isolated from the *M. oleifera* ethyl acetate fraction. Intravenous administration of any of these compounds at doses of 1 to 10 mg/kg resulted in hypotensive and bradycardiac actions in anesthetized rats, possibly mediated through calcium antagonist effects [47]. In vivo activity showed that isothiocyanate glycosides and thiocarbamate were responsible for this robust hypotensive activity [80]. 

Acetylcholine Esterase (AchE) is an enzyme widely distributed in neuromuscular junctions and the brain cholinergic synapses. Its principal biological activity in cholinergic synapses is to prevent the transmission of impulses through the rapid decomposition (hydrolysis) of acetylcholine (Ach) to acetate and choline [81]. Extracts from *M. oleifera* leaves and seeds were shown to reduce the activity of AchE, probably because of their phenolic constituents [77]. Phenolics are an important class of phytochemicals. Because of their (poly) hydroxyl groups, especially the 3′OH and 4′OH of their three-carbon chain, phenolics can donate electrons, therefore terminating the chain reaction process [77]. Structural similarity exists between naturally occurring polyphenols and the inhibitors of cholinesterase in terms of the hydrophobic component, molecular weight, and phenolic rings [82].

No biological effects were proven to be mediated through guanylyl cyclase activation. This fact leads to increased cyclic guanosine monophosphate (cGMP) synthesis, which, in turn, activates specific proteins, resulting in different actions, including smooth muscle relaxation, cardiac protection, neuronal plasticity, and endothelial permeability [83]. The cGMP activities were found to be terminated by the enzyme phosphodiesterase 5 (PDE-5) [83]. *M. oleifera* extracts decreased PDE-5 activity [77]. Previous reports revealed the capacity of medicinal plant extracts, especially those with high flavonoid contents, to inhibit the activity of PDE-5 [84]. The potential of *M. oleifera* extract to inhibit PDE-5 action could therefore be associated with its essential flavonoid content [14,84].

One of the primary critical regulators of hypertension is the renin–angiotensin system (RAS). It exhibits its anti-hypertensive actions mainly through the vasoactive peptide angiotensin II, released under angiotensin-converting enzyme (ACE) action following a blood pressure increase [85]. Inhibition of ACE has hypotensive effects. Diets supplemented with extracts from Moringa oleifera seed and leaves in rats reduced ACE activity [77]. There is evidence that phenolic compounds exhibit ACE inhibitory actions through the establishment of hydrogen bonds and hydrophobic interactions with the hydrophobic enzyme active site [86]. 

In addition, the anti-atherosclerotic and hypolipidemic effects of *M. oleifera* leaves were also shown in a different study [87,88]. Atherosclerosis, a highly chronic inflammatory disease, is closely associated with an increase in serum malondialdehyde (MDA). An increase in serum MDA suggests an increase in oxygen radical levels. Thus, the endothelial cell injury represents a critical initial event in atherosclerosis pathogenesis. Atherosclerosis pathogenesis begins with the ‘fatty streak’ lesions (accumulation of excess cholesterol and cholesteryl esters) in macrophage ‘foam’ cells within the intima of arteries [89,90]. In the pathogenesis of atherosclerosis, lipid accumulation is followed by chronic inflammation of the major arteries at some susceptible sites in their walls [90]. This chronic inflammation could result in ‘fatty streaks’ and then evolve into fibrous plaques [89]. The rupture of the plaque is performed through the action of enzymes released by activated macrophages. Once the plaque ruptures, the content is exposed to blood and could finally result in thrombosis. This thrombosis, therefore, may modify the shape of plaque and occlude the blood vessel lumen. The final results of stenosis provoked by the plaques are acute coronary syndrome, fatal arrhythmias, myocardial infarction, and sudden cardiac death [89,90]. The extract from the leaves of *M. oleifera* significantly prevented atherosclerotic plaque formation and development in the internal carotid of rabbits nourished with a diet highly supplemented with cholesterol. Interestingly, the capacity of the extract to prevent the formation of atherosclerotic plaque was highly comparable to that of simvastatin, the oral antilipemic agent that belongs to the statin class of medications and is largely used to manage abnormal lipid levels by inhibiting the endogenous liver production of cholesterol [88,89].

#### 3.3.2. Type 2 Diabetes: Chronic Inflammatory Disease and *M. oleifera*

Diabetes is a metabolic disease mainly manifesting through chronic hyperglycemia, resulting from impairment in insulin secretion and/or insulin action, with severe consequences [90]. Symptoms of elevated blood sugar are, among others: frequent micturition, increased thirst, and hunger. Hyperglycemia in diabetes is followed by impairment in lipids, carbohydrates, and protein metabolism [91]. Without any treatment, diabetes may lead to many complications: cardiovascular disease, diabetic ketoacidosis, hyperosmolar hyperglycemic state, foot ulcers, stroke, chronic kidney disease, eye damage, or death [92]. Type 2 diabetes is traditionally characterized by insulin resistance/reduced systemic insulin sensitivity, and islet β-cell dysfunction [93]. Chronic tissue inflammation is the key contributing factor to type 2 diabetes [93]. Elevated glucose and lipid levels, particularly saturated fatty acids, are hallmarks of insulin resistance and synergistically increase FAS expression within the cell [94]. This fact contributes to diabetes type 2 pathogenesis via endoplasmic reticulum stress and the subsequent generation of reactive oxygen species. Both events culminate and induce pro-inflammatory cytokine production [94]. In particular, IL1b secretion has been known as the mediator of β-cell dysfunction and death, and its effects are potentiated by interferon c (IFNc) and TNF-α. Inflammatory cytokines act on an inhibitor of the kappa light polypeptide gene enhancer in b-cells (IKKB) and on mitogen-activated protein kinase 8/JNK1 to inhibit insulin action directly via serine phosphorylation of substrates one and two of the insulin receptor [94]. 

In addition, in streptozotocin-induced diabetic rats, after 21 treatment days with *M. oleifera* aqueous leaf extract, the blood glucose level decreased. When diabetic and non-treated animals were compared to the control group, levels of organ damage markers differed significantly (*p* values: 0.0001). Treatment with the *M. oleifera* extract significantly reduced oxidative stress markers (hydrogen peroxide, MDA, and protein carbonyl) in the kidney, heart, and liver. Antioxidants in the diabetic non-treated group were reduced, while an increase in the group treated with *Moringa* extract was observed. From the pancreas and liver histologies, varied levels of inflammatory cell infiltration were observed, along with congestion and necrotic lesions. These tissue lesions were mild in *Moringa*-treated groups [95]. Furthermore, *M. oleifera* extract caused upregulation of glucose transporter 4 (GLUT 4), which is relevant in reversing insulin resistance in a similar way to pioglitazone, a standard antidiabetic agent [95]. GLUT 4 was downregulated in the untreated diabetic group compared to *Moringa*-treated groups, in which it was well expressed. Niazirin is a phenolic glycoside isolated from *M. oleifera* seed. Reports indicated that it could improve insulin resistance, hyperglycemia, hyperlipidemia, and non-alcohol fatty liver disease [96]. The significant biological effects of niazirin were shown to be mediated by its capacity to reduce lipid accumulation and gluconeogenesis and its capacity to improve lipid oxidation and glycolysis. Niazirin maintains energy homeostasis via the activation of the adenosine monophosphate-activated protein kinase (AMPK) signal pathway [96]. According to pharmacological and genetic investigations, AMPK is essential in maintaining glucose homeostasis [96]. The phosphorylation of its α-subunit induces AMPK activation. Activated AMPK then regulates its downstream targets, SirT1 and PFKFB3, and finally ameliorates glucose metabolism. Niazirin-induced phosphorylation of AMPKα and PFKFB3 in the db/db mice liver [96]. Other studies have shown that phosphorylation of AMPKα can activate SirT1, and SirT1 can then interact with PGC-1α to deacetylate it [97]. However, after niazirin treatment, SirT1 and PGC-1α expressions were almost brought to normal levels [96]. Insulin resistance in metabolic syndrome may result from the high secretion of TNF-α and the low secretion of IL-10 [96,98]. Inadequate secretion of cytokines such as TNF-α and IL-10 strengthens insulin resistance in db/db mice, resulting in adipocytes’ increased insensitivity to insulin [98]. After 4 treatment weeks with niazirin, in db/db mice, pro-inflammatory cytokine levels decreased, hyperglycemia and insulin resistance were alleviated, lipid metabolism was brought back to normal, and lipotoxicity was reduced [96].

Many antidiabetic plants are rich in phenolic compounds [99,100]. The antidiabetic properties of phenolic compounds may include, among others: inhibition of glucose metabolism enzymes, like α-glucosidase, α-amylase, and aldolase reductase; inhibition of insulin sensitization; induction of insulin-like glucose transport into adipocytes; inhibition of gluconeogenesis; increased GLP-1 receptor binding; insulin secretagogue activity; PPAR-γ-agonist; insulin-like activity; and an insulinotropic effect [100,101]. Some phenols of plant origin promote insulin secretagogue activity in β-cells via ATP-dependent K-channels or insulin–mimetic mechanisms. In contrast, others are essentially insulinotropic [101]. 

Moreover, phytofabrication of selenium nanoparticles with *M. oleifera* (MO-SeNPs) exhibited encouraging antidiabetic characteristics, displaying inhibition of alpha-amylase (ranging from 26.7% to 44.53%) and inhibition of the alpha-glucosidase enzyme (ranging from 4.73% to 19.26%), with the degree of inhibition being dependent on the dosage [102].

Also, in a clinical study, *M. oleifera* ameliorated the plasma lipid and glucose levels of type 2 diabetic subjects [103].

#### 3.3.3. Inflammatory Bowel Disease and *M. oleifera*

The intestine is the body’s largest digestive organ and is critical for digestion and nutrient absorption [104]. In normal physiological conditions, intestinal epithelial cells undergo selective permeation, allowing the passage of nutrients while preventing harmful substances from invading the intestinal epithelial cells [104,105]. The intestine is home to a group of microorganisms known as the “microbiota” [105]. The microbiota is a metabolically and immunologically complex active ecosystem composed of hundreds of thousands of microorganisms (viruses, bacteria, and some eukaryotes) that invade and colonize the digestive tract [105,106]. There is a dynamic relationship of mutual profits (symbiosis) established between the microbiota and human organism, which contributes to regular metabolic, immunological, and motor function maintenance, as well as to correct and adequate digestion and nutrient absorption [106,107]. The imbalance between the microbiota and gut defense system may result in aberrant inflammatory responses leading to neutrophil infiltration and chronic intestinal inflammation, as it is in the case of inflammatory bowel disease (IBD) [3,108]. IBD constitutes an immunological, histopathologically, and sometimes genetically heterogeneous group of bowel inflammation disorders, including Crohn’s disease (CD) and ulcerative colitis (UC) [3,108]. UC is a chronic illness at the origin of inflammation and/or ulceration of the large intestine’s (colon and rectum) epithelial lining, whereas Crohn’s disease can affect the layers of the alimentary tract and may even skip segments [109]. Their clinical manifestations include frequent diarrhea episodes, bloody feces, abdominal cramps, and over medium- to long-term weight loss. CD closely resembles UC [110]. Pathophysiological events associated with IBD are, among others, increases in specific pro-inflammatory mediators such as TNF-α, IL-1β, and IL-6 [109]; increased oxidative stress; impaired mucosa glycosaminoglycan (GAG) content; reduced short-chain fatty acid oxidation; increased permeability in the intestine; high sulfide synthesis; and reduced methylation [111]. 

Moreover, Kim et al. (2017) showed that, on dextran sulfate sodium (DSS)-induced acute and chronic UC, *M. oleifera* seed extract reduced colitis severity by attenuating the disease activity index (DAI) scores, increasing the colon lengths, and decreasing the colon weight/length ratios [111]. Furthermore, the extract also reduced the histopathological scores and colonic damage in acute UC. It decreased pro-inflammatory cytokines (myeloperoxidase (MPO), nitric oxide (NO), and TNF-α secretion) in the colon during acute and chronic colitis [112]. In acute UC, *M. oleifera* seed extract treatment was shown to reduce fecal lipocalin-2, downregulated gene expression of pro-inflammatory interleukin (IL)-1, IL-6, TNF-α, and inducible iNOS; upregulated claudin-1 and ZO-1 expression in acute and chronic colitis; as well as in chronic UC, upregulated GSTP1, which is an Nrf2 key mediator of phase II detoxifying enzyme [112]. Investigations by Noubissi et al. 2022 using *M. oleifera* leaf-aqueous extract on acetic acid-induced acute UC corroborate these findings [3].

Kaempferol (3,5,7-trihydroxy-2-(4-hydroxyphenyl)-4H-1-benzopyran-4-one is a flavonoid compound isolated from *M. oleifera*. Numerous preclinical investigations presented kaempferol (**10**) and some of its numerous glycosides as exhibiting a wide range of biological activities, including antioxidant and anti-inflammatory [113]. Feeding kaempferol (**10**) (0.1% to 0.3%) showed an effective decrease in the severity of colitis in DSS-induced colitis in mice. At 0.3%, kaempferol (**10**) decreased the plasma leukotriene B4 [LTB(4)] level in all treated animals, while NO and PGE2 contents decreased significantly [110]. Kaempferol (**10**) also suppressed MPO activity in the colon mucosa [114]. Additionally, in kaempferol (**10**) pre-treated animals, the quantity of TFF3 (a marker of goblet cell function) mRNA was upregulated, indicating its usefulness [110,115]. 

Astragalin, another flavonoid compound from *M. oleifera*, as well as KETTTIVR, an active peptide isolated from *M. oleifera* seeds, prevented weight loss, reduced the disease activity index, prevented colon shortening, and improved colon-damaging tissue in colitis mice [116,117]. Peng et al., 2020, also found that Astragalin reduced pro-inflammatory cytokines (such as TNF-α, IL-6, and IL-1β), and their related mRNA expression prevented macrophages and neutrophils colonic infiltration and ameliorated mucosal barrier function in the intestine [116]. They also revealed through Western blot analysis that Astragalin downregulated the NF-κB signaling pathway. Moreover, these authors showed that Astragalin or KETTTIVR partially reversed the gut microbiota alterations in colitis mice, mainly by increasing the potentially beneficial bacteria load (such as Ruminococcaceae) and decreasing the potentially harmful bacteria load (such as *Escherichia-Shigella*) [116], or by remodeling the intestinal mucosal barrier through inhibiting the JAK–STAT activation in colitis [117].

#### 3.3.4. *M. oleifera* Potential Effects on Non-Alcoholic Fatty Liver Disease (NAFLD)

Non-alcoholic fatty liver disease (NAFLD) is a global public health concern. The general term encompasses two subsets of patients [118]: individuals with non-alcoholic fatty liver (NAFL), with at least 5% hepatic steatosis without evidence of hepatocellular injury, and individuals with non-alcoholic steatohepatitis (NASH), defined by the presence of at least 5% hepatic steatosis and inflammation with hepatocellular injury, with or without fibrosis [118,119]. Non-alcoholic fatty liver disease (NAFLD) is associated with several metabolic risk factors, such as dyslipidemia, obesity, and type 2 diabetes mellitus, in many cases involving genetic predisposition [118,120]. Thus, NAFLD exists in two forms: simple steatosis, or NAFL, and non-alcoholic steatohepatitis (NASH) [118,121]. In NAFL (which is usually considered benign and reversible, with minimal risk of progression to cirrhosis or liver failure), there is a degree of hepatic steatosis without significant inflammation leading to hepatocellular injury or fibrosis [122,123]. NASH, in contrast, refers to hepatic inflammation and injury (with steatosis), which results in cellular necrosis [123]. The risk of NASH progression to cirrhosis and/or liver failure and hepatocellular carcinoma is increased [122,123]. Non-alcoholic fatty liver disease is usually caused by an imbalance in lipid acquisition (fatty acid uptake and de novo lipogenesis) and lipid removal (mitochondrial fatty acid oxidation) [124,125]. This leads to an increase in adipose tissue mass and, thus, to overweight and obesity [125]. The primary storage site for energy in the form of triglycerides (TG) is adipose tissue. It represents an important endocrine organ secreting hormones, cytokines, and chemokines called adipokines [122]. In obesity, enlarged adipose tissue experiences a dysregulation of adipokine production. The level of pro-inflammatory chemokines and cytokines such as monocyte chemotactic protein (MCP)-1, TNF-α, interleukin (IL)-6, and IL-8 is increased. These increased cytokines have been associated with insulin resistance [124,126].

Furthermore, free fatty acid infiltration in obesity leads to adipose tissue dysfunction. Increased levels of hepatic free fatty acids lead to increased lipid synthesis and gluconeogenesis [127,128]. Increased hepatic free fatty acids lead to peripheral insulin resistance, contributing to inflammation by serving as ligands for Toll-like receptor (TLR) 4 and inducing cytokine production, thereby contributing to NAFLD [122,129]. Hepatic fat accumulation can also result in oxidative stress, which ultimately leads to free oxygen radicals (ROS), which cause damage by oxidation of the cell components [130]. At high concentrations, ROS causes oxidative modifications of cellular macromolecules (DNA, lipids, proteins, and others), while the accumulation of these damaged macromolecules will induce liver injury [131,132]. 

High fat and sugar accumulation has been associated with opioid and dopamine receptor activation in the nucleus accumbens, the brain area controlling the development of cravings [121,133]. Furthermore, fructose increases blood flow to the brain region responsible for motivation and reward, failing to reduce satiety and contributing to obesity in NAFLD [121]. This activation of reward centers in response to certain macronutrients is coupled to the systemic reduction of glucagon-like peptide 1 (GLP-1) and the increase in ghrelin, both gut-derived hormones that promote satiety and stimulate hunger [133]. As a result of these changes, circulating triglyceride levels increase, which has implications for pathogenesis [134]. Leptin and adiponectin, adipose-derived hormones, are also implicated in NAFLD pathogenesis [135,136]. Leptin acts primarily and centrally to inhibit food intake and stimulate energy expenditure [135]. Adiponectin increases hepatic insulin sensitivity and reduces body fat [136]. It has been demonstrated that NAFLD patients have low adiponectin levels and are resistant to leptin, no matter their high leptin level [121]. In addition, leptin can promote stellate cell fibrogenesis by stimulating the expression of fibrogenic genes and inflammation in T cells [124].

Almatrafi et al. investigated the activities of *M. oleifera* leaf extract on the guinea pig hepatic steatosis model. In the control animals, they observed an accumulation of lipids while the *M. oleifera* treatment dose-dependently reduced cholesterol and TG levels and prevented the development of a steatotic phenotype. This fact could potentially be under the action of bioactive compounds present in *M. oleifera*, such as quercetin (**14**) and chlorogenic acid (**19**) (CGA), known to alter gene expression of major regulators of hepatic cholesterol and triglyceride synthesis and uptake [137]. In another investigation, CGA significantly inhibited fatty acid synthase activities, 3-hydroxy-3-methylglutaryl CoA reductase, and acyl-CoA cholesterol acyltransferase; it increased fatty acid β-oxidation and PPARα expression in mouse livers compared to a control group [134]. Lipid biosynthesis is regulated at the level of transcription by SREBP-1 and SREBP-2. SREBP-1c coordinates and controls fatty acids and TG synthesis, while SREBP-2 controls cholesterol. *M. oleifera* decreased SREBP-1c expression in guinea pigs [137]. In other studies, CGA decreased TG via reductions in SREBP-1c [138]. *M. oleifera* leaf biological activities were assessed in high fat diet-induced obese mice for 12 weeks [139]. Waterman et al. found a reduction of pro-inflammatory cytokines, IL-6, and IL-1β gene expression in the ileum and liver tissues of *M. oleifera* treated mice. *M. oleifera* CGA may have contributed to such effects since it was shown to suppress inflammatory cytokine transcription and inhibit NF-κB signaling pathway activation [139]. Histological evaluation of livers also demonstrated less lipid droplet accumulation in *M. oleifera* treated groups [139]. 

The ability of *M. oleifera* to decrease hepatic TG could be associated with its DGAT2 expression-lowering capacity. DGAT2 is one of the key enzymes from the endoplasmic reticulum involved in TG biosynthesis. It acylates at the sn-3 position the diacylglycerol, using fatty acyl CoAs [137]. Quercetin (**14**), a flavonoid abundantly found in *M. oleifera*, has been shown to prevent TG synthesis in Caco-2 cells, partly through inhibition of DGAT2 action [137]. Quercetin (**14**) reduced liver fat storage and serum lipid profiles via an alteration of the expression of genes related to fat metabolism. This effect is mediated by reduced expression of the peroxisome proliferator-activated receptor-alpha (PPARα) gene, increased expression of a liver gene associated with lipid metabolism, and a reduction in non-esterified fatty acid levels [130]. In addition, Joung et al. [140] found that fermented *M. oleifera* leaf extract upregulated the expression of genes related to fatty acid uptake (CD36), fatty acid β-oxidation (ACOX1), and lipolysis (ATGL and HSL). AMPK25 acts like an energy sensor, and it has been considered the therapeutic target in NAFLD and associated metabolic diseases. Fermented *M. oleifera* leaf extract increased AMPK phosphorylation. Greater AMPK activation, decreased lipogenesis, and increased lipolysis by fermented *M. oleifera* may decrease hepatic lipid accumulation. Thus, fermented *M. oleifera* supplementation may slow NAFLD progression [140]. 

In addition, Asgari-Kafrani et al. (2020) [130] identified three phenolic compounds: caffeic acid (**18**), quercetin (**14**), and gallic acid (**21**) in the leaves and stem of *M. oleifera* extracts. They investigated the antioxidant status of *M. oleifera* in NAFLD rats. They concluded that this plant exhibits its antioxidant activities in NAFLD rats by reducing total cholesterol, triglyceride, low-density lipoprotein, very low-density lipoprotein, alanine aminotransferase, and aspartate aminotransferase. The gallic acid (**21**) and caffeic acid (**18**) reported in *M. oleifera* may have directly contributed to the observed antioxidative effect. Hydroxyl groups in ring B and the 3-OH group are related to the superoxide-scavenging activity of flavonoids [130]. Various investigations have demonstrated that quercetin (**14**) decreases lipogenesis-associated genes [130]. Quercetin (**14**) ameliorates steatosis through an increase in fatty acid oxidation [130]. 

In another study, Bao et al. investigated niazirin’s (a bioactive compound isolated from *M. oleifera* seed) effects as well as its mechanisms of action on metabolic syndrome in db/db diabetic mice. They found that niazirin treatment reduced in the liver of db/db mice the two gluconeogenic enzymes’ abnormally intense activity, namely, PEPCK and G6Pase [96]. They realized that, after a four week niazirin treatment, they observed a reduction in liver cell swelling, inhibition of lipid accumulation in the liver, a decrease in inflammatory cells, and restoration of hepatocyte tissue architecture. Additionally, the observation of histology scores for NAFLD activity indicated a significant restructuring of the liver histology in db/db mice due to niazirin. They also found that niazirin significantly reduced lipid profile area percentage in the liver, indicating that niazirin treatment significantly ameliorated NAFLD in db/db mice. Thus, niazirin could improve liver steatosis and NAFLD [96].

## 4. Conclusions

Chronic inflammation contributes to the pathophysiology of numerous diseases, including cardiovascular diseases (atherosclerosis and hypertension), type 2 diabetes, enterocolitis, and NAFLD. Nowadays, the most commonly used drugs in modern medicine, including anti-inflammatory drugs, are from plant sources and are regarded as safe. The plant kingdom, therefore, represents an alternative to conventional drugs. In this overview, we present the pathophysiology of a certain number of chronic inflammation-induced diseases. We also reported the biological activities of *M. oleifera* and the chemical compounds underlying these observed activities, not forgetting their potential mechanisms of action. Taken together, the knowledge on *M. oleifera* phytochemical compounds we gathered in this review will provide a guide concerning inflammatory disease management. The regulatory effects of *M. oleifera* phytochemical compounds on the various steps of inflammation and different metabolic pathways were highlighted, and they may serve as therapeutic targets. Although there has been recent progress towards understanding the mechanisms underlying the diverse bioactivities of the *M. oleifera* plant, further studies are required to establish and confirm these activities firmly. This research could prove valuable for exploring the pharmacological potential of *M. oleifera* and contributing to the prospects of developing effective medicines for the benefit of human health.

## Figures and Tables

**Figure 2 plants-13-00020-f002:**
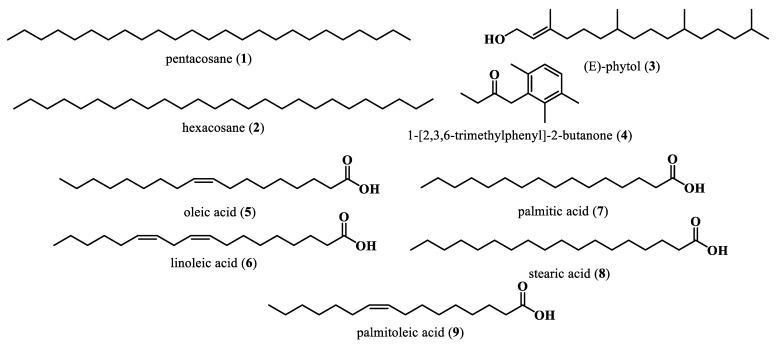
Volatile compounds and fatty acids from *M. oleifera*.

**Figure 3 plants-13-00020-f003:**
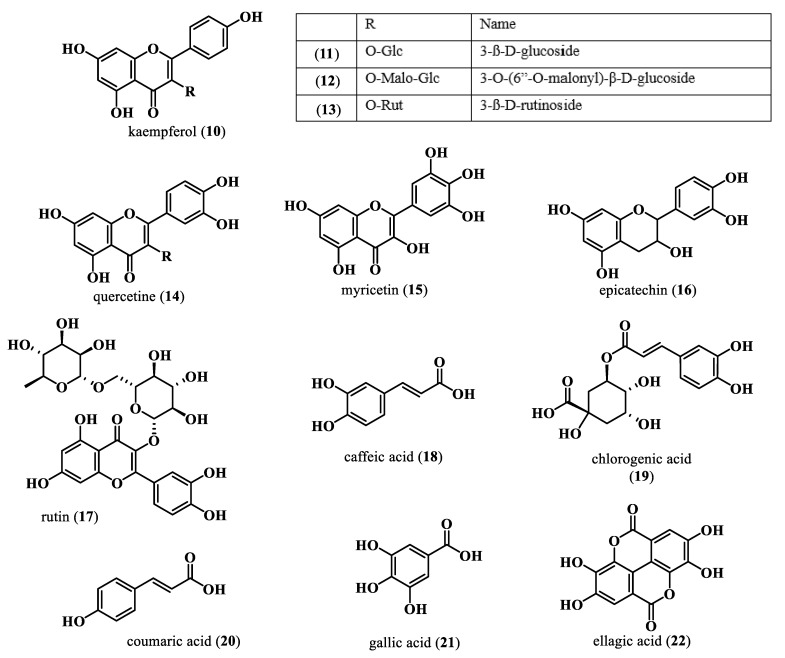
Polyphenolic compounds from *M. oleifera*.

**Figure 4 plants-13-00020-f004:**
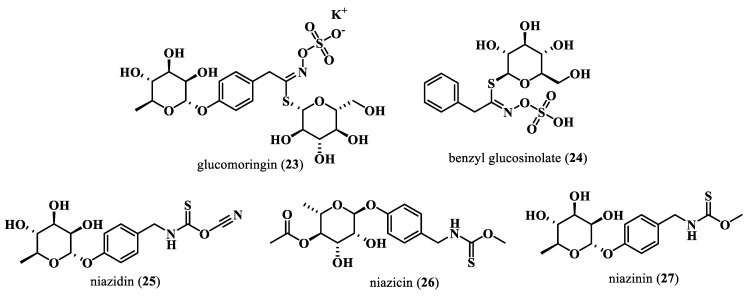
Sulfur compounds from *M. oleifera*.

**Figure 5 plants-13-00020-f005:**
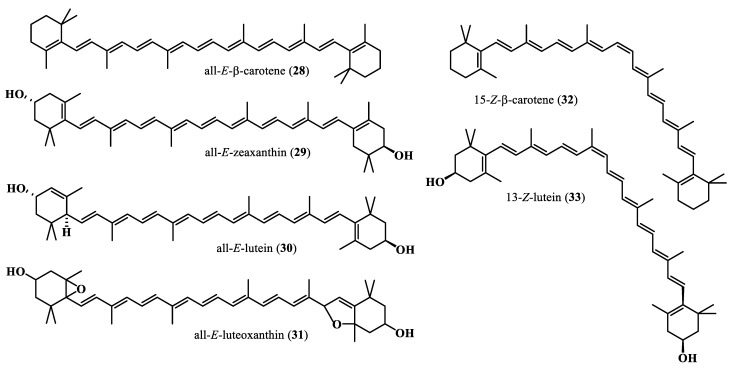
Carotenoidic compounds from *M. oleifera*.

**Figure 6 plants-13-00020-f006:**
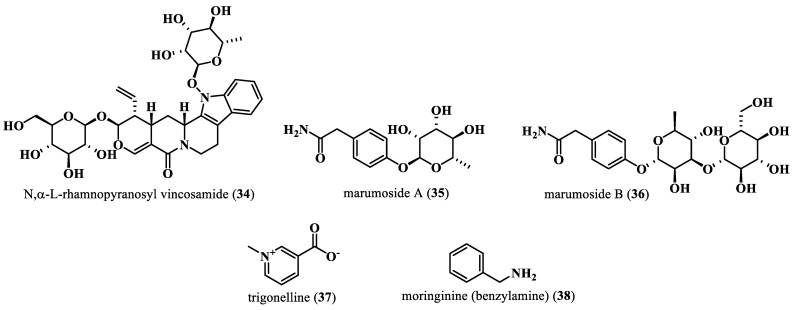
Alkaloids isolated from *M. oleifera*.

**Figure 7 plants-13-00020-f007:**
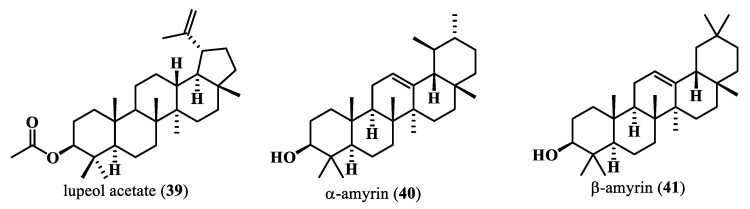
Terpenoids isolated from *M. oleifera*.

**Table 1 plants-13-00020-t001:** Bioactive compounds from *M. oleifera*, their bioactivity and therapeutic characteristics (in vivo and in vitro activities).

Part Used/Type of Extract	Experimental Model	Model of Study	Duration of the Treatment	Dosage	Control	Extracts/Secondary Metabolites	Bioactivity/Therapeutic Characteristics	Ref.
Leaves, fruits, and seeds/aqueous extracts	The experiment investigated the capacity to restrain oxidative DNA damage, antioxidant, and anti-quorum sensing (QS) capabilities		-	5–10 µg/mL 50 mg/mL	Positive control: furanone and garlic extract	Polyphenolic compounds (aqueous extracts)	Inhibition of proteins and lipid oxidation Antioxidant activities: nitric oxide and superoxide anion scavenging activities	[14]
Leaves	The study screened the functional metabolites of *M. oleifera* to evaluate their possible role in nerve regeneration after injury	in vivo Male albino mice	12 days	2 g/kg bw	Control: standard diet	Leaf extracts in n-Hexane, dichloromethane, ethyl acetate, ethanol, and methanol	Nerve regeneration (restoration of sensory and motor function) Oxidative stress management	[7]
Defatted seed flour	The study assessed the antioxidant and antibacterial activities of free and bound phenolic extracts	in vitro *B. cereus* *S. aureus* *E. coli* *Y. enterocolitica*	-	0.5–15.0 mg in 150 µL 0.5–1.5 mg/mL	Positive control: ascorbic acid	Phenolic compounds (extracts in ethanol, methanol, acetone, hexane, and chloroform)	Antioxidant, antimicrobial activities	[15]
Seeds	The experiments evaluated the antibacterial and antifungal activities of secondary metabolites	in vitro *Bacteria:* *S. aureus* *S. epidermidis* *E. coli* *E. aerogenes* *K. pneumonia* *P. aeruginosa* *B. subtilis* Fungal strains: *C. albicans* *T. rubrum* *E. floccosum*	-	10 mg/mL 1 mg/mL 100 µg/mL 10 µg/mL	Positive control: Ofloxacin and Clotrimazole Negative control: sterile distilled water	Glucosinolates (secondary metabolites extracts in acetone in CH_2_Cl_2_)	Antimicrobial activities	[16]
Seeds	The study evaluated the antitumoral activity of eight isolate compounds from *M. oleifera*	in vitro *EBV genome carrying lymphoblastoid cell* in vivo Specific pathogen-free female ICR mice	- 20 weeks	100, 10, 1, and 0.1 µg/mL 85 nmol in 0.1 mL acetone	Positive control: n-butyric acid Negative control: 12-O-tetradecanoyl-phorbol-13-acetate (TPA)	Glucosinolates, isothiocyanates, and sterols	Antitumor promoting activities	[17]
Seeds	The experiments assessed the antimicrobial activity of a polymeric, naturally extracted *M. oleifera* oil bionanocomposite film enriched with silver nanoparticles	in vitro *S. aureus,* *E. coli* *K. pneumoniae* *S. typhi* *P. aeruginosa* *S. flexneri* *C. albicans*	-	5–10 wt%	Ciprofloxacin and Fluconazole	Fatty acids, sterols, alkanes, and alcohol compounds	Antibacterial activity	[18]
Seeds	The study assessed the cytotoxic activity of seed essential oil obtained from *M*. *oleifera*	in vitro HeLa, HepG2, MCF-7, CACO-2, and L929 cell lines	24 h	0.15 to 1.0 mg/mL	DMSO	Essential oils (extracted from seeds through cold pressing)	Antiproliferative activity	[19]
Flowers		PC3 cell lines		0.01–100 µg/mL	DMSO	Methanol extracts	Anticancer activity	[20]
Leaves	The study has investigated the anticancer activity of the *M. oleifera* leaf extract	in vitro murine Non-Hodgkin Lymphoma (NHL) in vivo Balb/c mice	24 h 16 days	100 to 450 µg/mL 100 and 200 mg/kg	Healthy mice without Dalton’s lymphoma cell transplantation	Methanol-based leaf extract	Triggers apoptosis and inhibits the growth of Dalton’s lymphoma	[21]
Seeds	Experiments assessed the antioxidant, antihypertensive, and potential cardioprotective properties of bioactive peptides	in vivo Wistar rats	- -	- 200 mg/kg	- -	Peptides obtained through enzymatic hydrolysis of *M. oleifera* seed (shorter peptides (1–3 kDa) and longer peptides (>10 kDa))	Oxidative stress management; antihypertensive, and cardioprotective properties	[22,23]
Seeds	The study compared the structural and functional properties of albumin and globulin in *M. oleifera* seeds with those of the isoelectric pH-precipitated protein isolate	-	-	6.25 mg/mL	-	Globulins, Albumins, Iso–electric precipitated isolates	Reduce free radicals Globulin-enhanced metal ion chelation activity Antioxidant properties	[24]
Seeds	The experiments compared the antioxidant and angiotensin-converting enzyme (ACE) inhibitory properties of *M. oleifera* seed protein isolate (ISO)	*-*	-	-	-	Protein hydrolysate fractions (<1 kDa, 3–5 kDa, and 5–10 kDa)	Antioxidative properties ACE inhibition;	[25]
Seeds	The study evaluated the biofunctional properties of total hydrolysates and peptide fractions from protein isolates of moringa seeds	in vitro ACE extracted from rabbit lungs	5 h	-	-	Protein hydrolysate fractions >10 kDa)	Antioxidant, antihypertensive, and antidiabetic properties;	[26]
Seeds	The study evaluated the nutritional composition of Indian *M. oleifera* seed, the antioxidant activity of its polypeptides, and the protective effects on H_2_O_2_ oxidative-damaged Chang liver cells	in vitro Chang liver cell line	3 h	100, 300, and 500 µM	PBS	Peptide isolates and hydrolysate Fractions (>3.5 kDa) PFE, GY, YTR, QY, FG, SF, IN, SP, YFE, IY, LY	Oxidative stress management Antioxidant activity Hepatoprotector	[27,28]
Seeds	Seed powder was extracted in hexane, petroleum ether, ethyl acetate, or methanol, and the study evaluated the extracts antimicrobial activity	in vitro *E. coli* *P. aeruginosa* *S. aureus* *C. cladosporioides* *P. sclerotigenum*	-	-	-	Glucosinolates and isothiocyanates	Flocculating and antimicrobial activities	[29]
Seeds	The experiments assessed the antimicrobial activities of isolated compounds from seed extracts of *M. oleifera* and their synergistic effect through a hybridized complex of organic–inorganic composite materials	in vitro *S. aureus* *E. coli* *P. aeruginosa* *C. albicans* *A. niger*	24 h (37 °C) 48 h (25 °C) 5 days (25 °C)	5 mg	-	Glucosinolates and isothiocyanates (ethanol, methanol, hexane, acetate ethanol extracts)	Antimicrobial activities	[30]
Seeds	The study evaluated the antimicrobial activities of 4-(α-l-rhamnosyloxy) benzyl glucosinolate isolated from *M. oleifera* seed and its protective effect on an experimental model of spinal cord injury	in vitro *S. aureus* *E. casseliflavus* *C. albicans* in vivo Male adult C57Bl/6 mice	24 h 8 days	- 10 mg/kg	PBS Gentamicin Chloramphenicol Negative control: naive group (no treatment) Positive control: GMG-ITC (control group)	Glucosinolates and isothiocyanates	Antibiotic activity Nerve regeneration	[31,32,33]
Leaves	The study evaluated the anxiolytic and anti-colitis effects of *M. oleifera* leaf-aqueous extract on acetic acid-induced colon inflammation in rat	in vivo Adult albino Wistar rats	20 days	25, 50, and 100 mg/kg	Loperamide	-	Anxiolytic, anti-inflammatory Antioxidant and anti-colitis properties	[3]
Seeds	The study evaluated the dietary isothiocyanate-enriched moringa seed extract on glucose tolerance in a high fat diet mouse model and its modulatory activity on the gut microbiome	in vivo Male Sprague–Dawley rats	12 weeks	0.54 and 0.73% of moringa seed extract containing moringa isothiocyanate-1	Vehicle control: 15% sodium carboxymethyl cellulose	Glucosinolates, phenolic glycosides, flavonoids, and carbohydrates	Anti-inflammatory Antioxidant properties Improves glucose tolerance and modulates the gut microbiome	[34,35]

Legend: DPPH: 2,2-diphenyl-1-picrylhydrazyl; HeLa: human cervical cancer; HepG2: human hepatocellular carcinoma; ACE: angiotensin-I converting enzyme; Amino acid nomenclature: I: isoleucine; S: serine; Y: tyrosine; F: phenylalanine; E: glutamic acid; N: asparagine; Q: glutamine; G: glycine; P: proline; L: leucine; T: threonine; R: arginine.

## Data Availability

Not applicable.

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
