# Peer review of "Bioactive Compounds in Moringa oleifera: Mechanisms of Action, Focus on Their Anti-Inflammatory Properties"

_plants, 2023, doi:10.3390/plants13010020_

Round 1
Reviewer 1 Report
Comments and Suggestions for Authors
The work's concept is relevant and well-structured.
This review, which details the benefits of moringa, is of scientific and societal relevance.
Given the aforementioned, I believe the work should be published currently written.
Author Response
Thank you very much for your evaluation and appreciations.
Reviewer 2 Report
Comments and Suggestions for Authors
The work is interesting, as it generates a review of the capacity of bioactive compounds present in Moringa (a product traditionally used in various cultures and widely used worldwide). The document has enough information about the inflammatory process associated with diseases, and how the consumption of Moringa (in different forms) could help counteract the process, and what types of compounds are the participants in it. However, for better understanding, it requires improving some minor details.
· In the abstract, it is not necessary to indicate that the search was carried out in Pubmed and Google.
· On line 39, phytochemicals, polyphenols, and flavonoids are indicated as if they were independent compounds. Improve that point.
· On line 45 the types of mechanisms are separated with a period, while before it is done with a semicolon (;). Review the same in the rest of the document (there are also variations).
· In the introduction, simplify from line 97 to 105, since the effect of light on the content of compounds in Moringa has no relevance for this work.
· Review the connection between paragraphs, both in the introduction and the rest of the text. They cannot remain independent paragraphs, but rather as part of a single document. Use connectors such as “In addition”, “Also”, “Additionally”, among others, which are simple but allow generating a connection between paragraphs.
· Review Table 1 completely and write in uniform language. In some cases, excessive details are explained, while in others only the indicated information is presented. For example, the information in reference 7 gives extensive information on the model, but does not really provide the experimental model: is it in vitro, in vivo? What do you mean by groups (animals, what type?). In addition, the following columns provide information on dosage, time of use, etc. On the other hand, the same thing happens with the therapeutic activity or action, where a large piece of writing is delivered, which may be simple: Nerve regeneration (restoration of sensory and motor function); oxidative stress management. While in other cases you only indicate “antioxidant capacity”. Write in the same way and with the same level of detail. Make it explanatory but simple.
· Review the title of figure 2, Not all the compounds presented in it are volatile.
· Review lines 275 and 276 and improve the wording.
· Review lines 339 to 342 and make a single sentence of all the information.
· Check the page numbering of the document carefully.
Author Response
Thank you very much for your evaluation and appreciations. We have carefully considered each of your suggestions and have made the necessary revisions to address the concerns raised:
- In the abstract, it is not necessary to indicate that the search was carried out in Pubmed and Google.
Thank you, we replaced the sentence.
- On line 39, phytochemicals, polyphenols, and flavonoids are indicated as if they were independent compounds. Improve that point.
Thank you, we replaced the sentence.
- On line 45 the types of mechanisms are separated with a period, while before it is done with a semicolon (;). Review the same in the rest of the document (there are also variations).
Thank you, we corrected the phrase.
- In the introduction, simplify from line 97 to 105, since the effect of light on the content of compounds in Moringa has no relevance for this work.
Thank you, we simplified the phrase.
- Review the connection between paragraphs, both in the introduction and the rest of the text. They cannot remain independent paragraphs, but rather as part of a single document. Use connectors such as “In addition”, “Also”, “Additionally”, among others, which are simple but allow generating a connection between paragraphs.
Thank you very much for your observation, we did it.
- Review Table 1 completely and write in uniform language. In some cases, excessive details are explained, while in others only the indicated information is presented. For example, the information in reference 7 gives extensive information on the model, but does not really provide the experimental model: is it in vitro, in vivo? What do you mean by groups (animals, what type?). In addition, the following columns provide information on dosage, time of use, etc. On the other hand, the same thing happens with the therapeutic activity or action, where a large piece of writing is delivered, which may be simple: Nerve regeneration (restoration of sensory and motor function); oxidative stress management. While in other cases you only indicate “antioxidant capacity”. Write in the same way and with the same level of detail. Make it explanatory but simple.
Thank you very much, we simplified and replaced the table.
- Review the title of figure 2, Not all the compounds presented in it are volatile.
Thank you very much, we corrected the legend of the figure.
- Review lines 275 and 276 and improve the wording.
Thank you very much, we have improved the text.
- Review lines 339 to 342 and make a single sentence of all the information.
Thank you very much, we replaced the sentence.
- Check the page numbering of the document carefully.
Thank you very much, we corrected the page number insertion in different sections.
Reviewer 3 Report
Comments and Suggestions for Authors
The manuscript refers to the compilation of information in relation to the anti-inflammatory mechanisms of Moringa oleifera. In general, the information provided in the manuscript is interesting, it is distributed appropriately and provides a collection of interesting and enriching information for readers; however, much of this information has already been reported previously in other reviews. The mechanism of anti-inflammatory activity of Moringa has the highest percentage of current documents, which is enriching, however, it is suggested, as far as possible, to only include the most current information 5 years to date since there are references prior to these years. same that have already been reported in various review works. Additionally, the similarity checker detects a high percentage of similarity, so it is suggested that a more detailed review of the topic be carried out to provide more up-to-date information as well as reduce the similarity percentage.
Author Response
Thank you very much for your evaluation.
We have updated several old references, but we kept a few of former references due to their contributions that we used. Concerning the similarities report, we checked again, and we reformulated several phrases, but some common terms cannot be replaced.
Round 2
Reviewer 3 Report
Comments and Suggestions for Authors
The authors of the manuscript entitled "Bioactive compounds in Moringa oleifera: mechanisms of action, focus on their anti-inflammatory properties", have addressed the observations and suggestions made in the paper. The added information has allowed us to obtain a manuscript that can be considered of interest to the readers of the journal. Additionally, they have added information that is current and that has not been presented in other reviews.